# Cyclic di-AMP traps proton-coupled K$^+$ transporters of the KUP family in an inward-occluded conformation

Michael F. Fuss [1,9], Jan-Philip Wieferig[2,9], Robin A. Corey[3,8,9], Yvonne Hellmich [1], Igor Tascón [1,5,6], Joana S. Sousa[2,7], Phillip J. Stansfeld [4], Janet Vonck [2] ✉ & Inga Hänelt [1] ✉

Cyclic di-AMP is the only known essential second messenger in bacteria and archaea, regulating different proteins indispensable for numerous physiological processes. In particular, it controls various potassium and osmolyte transporters involved in osmoregulation. In *Bacillus subtilis*, the K$^+$/H$^+$ symporter KimA of the KUP family is inactivated by c-di-AMP. KimA sustains survival at potassium limitation at low external pH by mediating potassium ion uptake. However, at elevated intracellular K$^+$ concentrations, further K$^+$ accumulation would be toxic. In this study, we reveal the molecular basis of how c-di-AMP binding inhibits KimA. We report cryo-EM structures of KimA with bound c-di-AMP in detergent solution and reconstituted in amphipols. By combining structural data with functional assays and molecular dynamics simulations we reveal how c-di-AMP modulates transport. We show that an intracellular loop in the transmembrane domain interacts with c-di-AMP bound to the adjacent cytosolic domain. This reduces the mobility of transmembrane helices at the cytosolic side of the K$^+$ binding site and therefore traps KimA in an inward-occluded conformation.

In bacteria and archaea, potassium ions are essential, playing roles in osmoregulation[1,2], pH homeostasis[3–5], regulation of protein synthesis[6], enzyme activation[7,8] such as the ribosome[9,10], membrane potential adjustment[11] and electrical signalling[12–14]. K$^+$ homeostasis must be strictly maintained as deviations and fluctuations of potassium levels have lethal consequences for the cell[15,16]. In Gram-positive bacteria like *Bacillus subtilis* the second messenger cyclic di-AMP (c-di-AMP) is essential for the regulation of channels and transporters that maintain potassium homeostasis[17]. At elevated intracellular K$^+$ concentrations, c-di-AMP is present at increased concentrations and regulates potassium-transporting proteins in two ways: Firstly, it directly binds its target proteins, leading to the inhibition of potassium uptake systems like K$^+$ channel KtrAB[18,19] and K$^+$/H$^+$ symporter KimA[17,20] or the activation of potassium exporters like CpaA[21] and KhtTU[22]; secondly, c-di-AMP suppresses gene expression by binding to the *ydaO* riboswitch[23] that among others controls the transcription of the *kimA* and *ktrAB* genes. The ability of c-di-AMP to modulate both the expression and the activity of the same proteins makes it a key player in potassium homeostasis[24]. However, while the control of gene expression is well understood, it remains unclear how c-di-AMP binding controls the activity of potassium transporters and channels.

[1]Institute of Biochemistry, Goethe University Frankfurt, Frankfurt am Main, Germany. [2]Department of Structural Biology, Max Planck Institute of Biophysics, Frankfurt am Main, Germany. [3]Department of Biochemistry, University of Oxford, Oxford, UK. [4]School of Life Sciences & Department of Chemistry, University of Warwick, Coventry CV4 7AL, UK. [5]Instituto Biofisika (UPV/EHU, CSIC), University of the Basque Country, Leioa, Spain. [6]Ikerbasque, Basque Foundation for Science, Bilbao, Spain. [7]UCB Pharma, UCB Biopharma UK, Slough SL1 3WE, UK. [8]Present address: School of Physiology, Pharmacology and Neuroscience, University of Bristol, Bristol, UK. [9]These authors contributed equally: Michael F. Fuss, Jan-Philip Wieferig, Robin A. Corey. ✉e-mail: janet.vonck@biophys.mpg.de; haenelt@biochem.uni-frankfurt.de

The c-di-AMP-sensitive, high-affinity $K^+/H^+$ symporter KimA from *B. subtilis* is particularly required for the uptake of potassium at low external potassium concentrations in acidic environments. KimA exploits the inward-directed proton gradient to accumulate $K^+$ against its concentration gradient. As a member of the amino acid-polyamine-organocation (APC) superfamily it has the classical LeuT-fold with the first ten transmembrane helices (TMHs) adopting a 5 + 5 inverted repeat. TMHs 11 and 12 connect the transmembrane domain (TMD) to the cytosolic domain (CD) of KimA, which consists of four alpha helices and a five-stranded beta sheet[25]. KimA is a homodimer, stabilised by the swapping of the cytosolic domains with respect to the transmembrane domain. A long loop connects the swapped cytosolic domain to the last helix of the TMD. The dimeric cytosolic domains adopt a fold similar to a phosphopantetheine adenylyltransferase (PPAT) domain and have been suggested to bind c-di-AMP[25,26]. However, no structural information is available for the c-di-AMP binding site and, consequently, the inhibition mechanism of potassium uptake by c-di-AMP remains elusive. In this study, we report structural and functional insights into the binding of c-di-AMP to KimA and how it inhibits the transporter.

## Results

### C-di-AMP binds to KimA in a cooperative manner and inhibits potassium uptake

The uptake of potassium ions through KimA is known to be impaired by c-di-AMP[16,20]. Here, we confirmed the hypothesis and showed that binding of c-di-AMP leads to a significant decrease of potassium uptake velocity under in vivo conditions. We co-produced KimA with the inactive diadenylate cyclase variant $CdaA_{D171N}$[27] in *Escherichia coli* LB2003 cells, a strain that lacks all endogenous potassium uptake systems[28], and determined potassium uptake into the potassium-depleted cells, obtaining a $V_{max}$ of $269.3 \pm 24.0$ nmol*mg dw$^{-1}$*min$^{-1}$ and a $K_m$ of $0.49 \pm 0.09$ mM. Upon the co-expression of KimA with active, c-di-AMP-synthesising CdaA, the $V_{max}$ was reduced by ~64% to $96.7 \pm 10.1$ nmol*mg dw$^{-1}$*min$^{-1}$, while the $K_m$ did not change ($0.66 \pm 0.31$ mM) (Fig. 1a, Table 1).

In our previous structural study[25], when c-di-AMP was added to KimA purified in styrene maleic-acid lipid particles (SMALPs) before EM grid preparation, binding of c-di-AMP to KimA was not observed. In agreement with this we show here that the melting temperature of detergent-purified KimA did not significantly change upon the titration of increasing concentrations of c-di-AMP in differential scanning fluorometry (DSF) measurements (Fig. 1b), suggesting that c-di-AMP did not bind in these conditions. To overcome this limitation, we co-produced KimA with the diadenylate cyclase CdaA in *E. coli* LB2003 cells, as this combination showed KimA inhibition in vivo, and then purified KimA from this condition. Interestingly, this sample already had a slightly increased melting temperature from $48.1 \pm 0.8$ °C to $50.7 \pm 1.9$ °C, prior to the addition of c-di-AMP. The addition of a three-fold molar excess of c-di-AMP led to a further 6.6 °C increase of the melting temperature to $54.7 \pm 0.7$ °C (Fig. 1b). Hence, during purification, a fraction of c-di-AMP appeared to remain bound to KimA, which enabled the binding of further

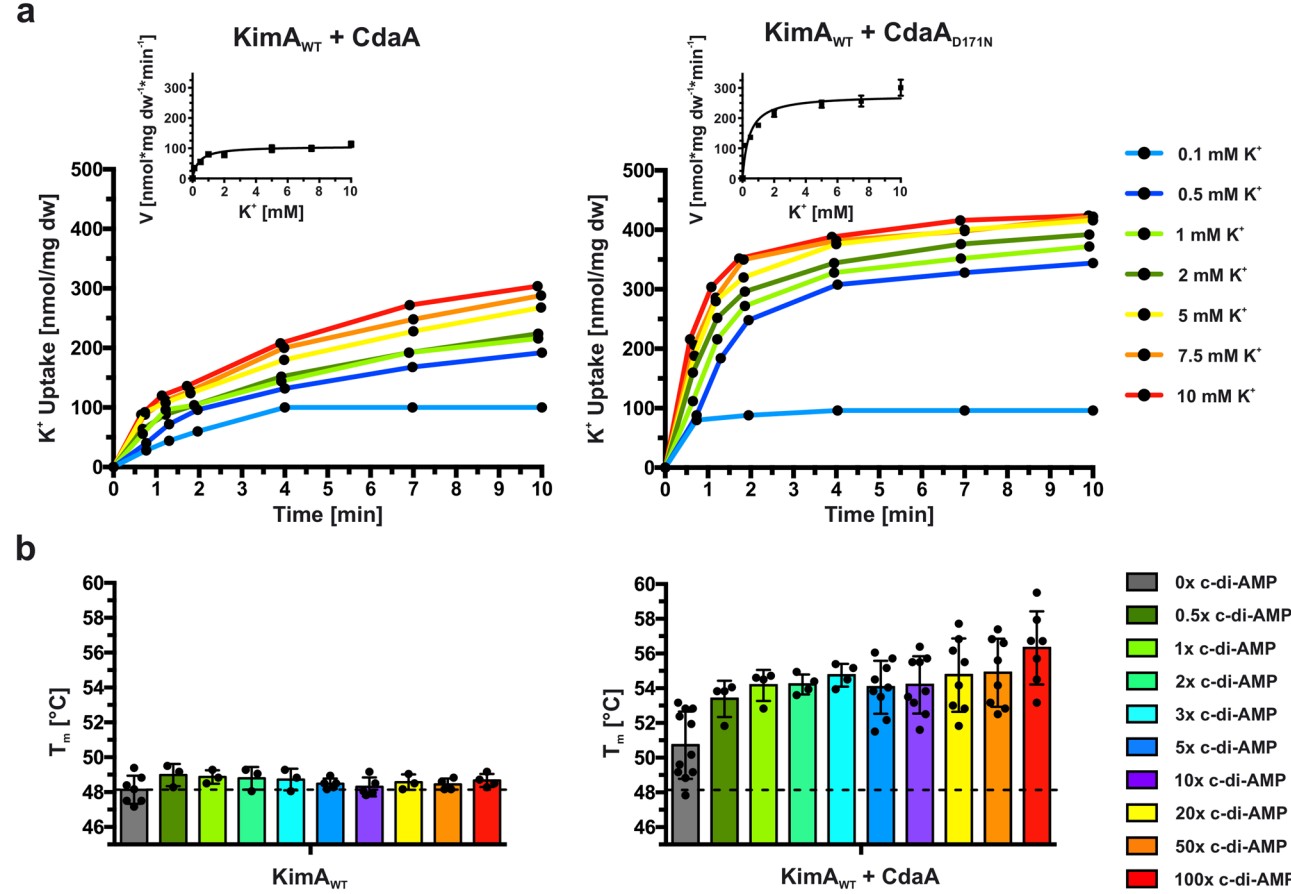

**Fig. 1 | The effect of c-di-AMP binding to KimA in vivo and in vitro. a** Whole-cell $K^+$ uptake assay in *E. coli* LB2003 cells producing KimA with active (CdaA) or inactive ($CdaA_{D171N}$) diadenylate cyclase. Michaelis-Menten plot shown in graph with mean value ± SER. Representative experiments shown (*n* = 3). **b** Melting temperatures of KimA purified from cells with CdaA present (KimA WT + CdaA) or absent (KimA WT) and incubated with an increasing c-di-AMP concentration given in x-fold molar excess over KimA. Determined with Differential Scanning Fluorometry (DSF). Dashed line indicates $T_m$ of KimA WT w/o c-di-AMP addition. Data point represents the average and error bars the standard deviation of measurements from at least biological triplicate (*n* ≥ 3 right and *n* ≥ 4 left).

c-di-AMP under ex vivo conditions. This suggests a cooperative behaviour of c-di-AMP binding to KimA at least under in vitro conditions.

**Structural characterisation of the binding of c-di-AMP to KimA**

To determine the binding site of c-di-AMP and the effect of c-di-AMP binding on the overall structure of KimA, cryo-EM specimens of purified KimA that had been co-expressed with *cdaA* were prepared in both DDM and amphipols. KimA in DDM was additionally incubated with a ten-fold molar excess of c-di-AMP after purification. 332k and 296k particles were processed to reconstruct cryo-EM maps at 3.3 Å and 3.8 Å resolution, respectively. The overall structure of KimA obtained from the preparation in DDM resembles the previous structure obtained in SMALPs[25] (RMSDs of TMD (residues 30–462): 1.31 Å, CD (residues 462–606): 1.65 Å, global RMSD: 1.74 Å). The two TMDs are tilted towards each other, forming a dimer interface at the

extracellular side that would enforce a bending of the membrane by ~130° against its natural curvature. As speculated for the preparation of KimA in SMALPs, the tilting likely was caused by a loss of lipids between the TMDs during purification. In contrast, the map of KimA in amphipols shows the upright dimer architecture of the TMDs (Supplementary Fig. 1a). At the extracellular side an elongated non-protein density is visible between both protomers, indicating that amphipol molecules wrapped around both TMDs individually. The structure resembles the upright dimer that was previously obtained following molecular dynamics (MD) simulations of KimA in a lipid bilayer[25], with an RMSD of only 3.4 Å (Supplementary Fig. 1b). The pivot point for the tilting of TMDs is located around the C-terminal end of TMH 12, which extends out of the membrane (residues 459–464). Otherwise, the architectures of the TMDs (RMSD 0.55 Å for residues 30–462) and of the cytosolic domains (RMSD 0.66 Å for residues 462–606) of KimA reconstituted in amphipols and in DDM are very similar (global RMSD 1.26 Å). Also, the dimer interfaces that are mainly formed by the long connecting loops (residues 462–474) and by a beta-sheet connecting the two cytosolic domains are not affected (Fig. 2a). A comparison of the resulting structures of KimA in DDM and in amphipols with the previously reported structure from SMALPs[25] suggests that all of them represent a similar inward-occluded conformation. In agreement with this, a strong density for the bound substrate potassium ion is observed in the 3.3 Å cryo-EM map (Fig. 2b, Supplementary Fig. 2a). The potassium ion is coordinated by Tyr43 (2.9 Å), the carbonyl and carboxyl of Asp36 (2.8 Å and 3.3 Å, respectively), hydroxyl and carbonyl of Thr230 (2.9 and 3.2 Å, respectively) and Tyr377 (3.2 Å). The map from KimA obtained in SMALPs suggested that Ser125 is also part of the K+ binding site, but in MD simulations this was not the case[25]. The higher resolution map shows that the distance between Ser125 and the potassium ion (4.9 Å) is too long for coordination (Fig. 2b), confirming the MD simulations. In contrast to the previous map, no extra densities are localized below Asp36 and Tyr377. Asp36 and Tyr377 are suggested to function as an intracellular gate. Potassium ions below them were hypothesized to hinder the opening of the gate by a trans-inhibition mechanism[25]. An explanation for the lack of the inhibitory ions could

**Table 1 | Kinetic parameters of KimA variants in the presence or absence of c-di-AMP (i.e., active (CdaA) or inactive (CdaA_{D171N}) diadenylate cyclase)**

| Variant | $V_{max}$ [nmol*mg dw$^{-1}$*min$^{-1}$] | $K_m$ [mM] |
|---|---|---|
| KimA_{WT} + CdaA | 96.7 ± 10.1 | 0.66 ± 0.31 |
| KimA_{WT} + CdaA_{D171N} | 269.3 ± 24.0 | 0.49 ± 0.09 |
| KimA_{R337A} + CdaA | 111.7 ± 25.5 | 0.54 ± 0.41 |
| KimA_{R337A} + CdaA_{D171N} | 198.9 ± 35.8 | 0.39 ± 0.08 |
| KimA_{Y118A} + CdaA | 210.6 ± 29.0 | 18.85 ± 5.82 |
| KimA_{Y118A} + CdaA_{D171N} | 202.8 ± 19.4 | 4.8 ± 0.75 |
| KimA_{N237A} + CdaA | 207.9 ± 23.6 | 7.80 ± 2.23 |
| KimA_{N237A} + CdaA_{D171N} | 161.9 ± 41.8 | 7.93 ± 4.21 |
| KimA_{A481WS582W} + CdaA | 194.2 ± 49.2 | 0.35 ± 0.15 |
| KimA_{A481WS582W} + CdaA_{D171N} | 182.0 ± 44.1 | 0.33 ± 0.08 |

Determined by fitting the Michaelis-Menten plot with Michaelis-Menten equation. Mean and standard deviation from three independent whole-cell potassium uptake assays are shown.

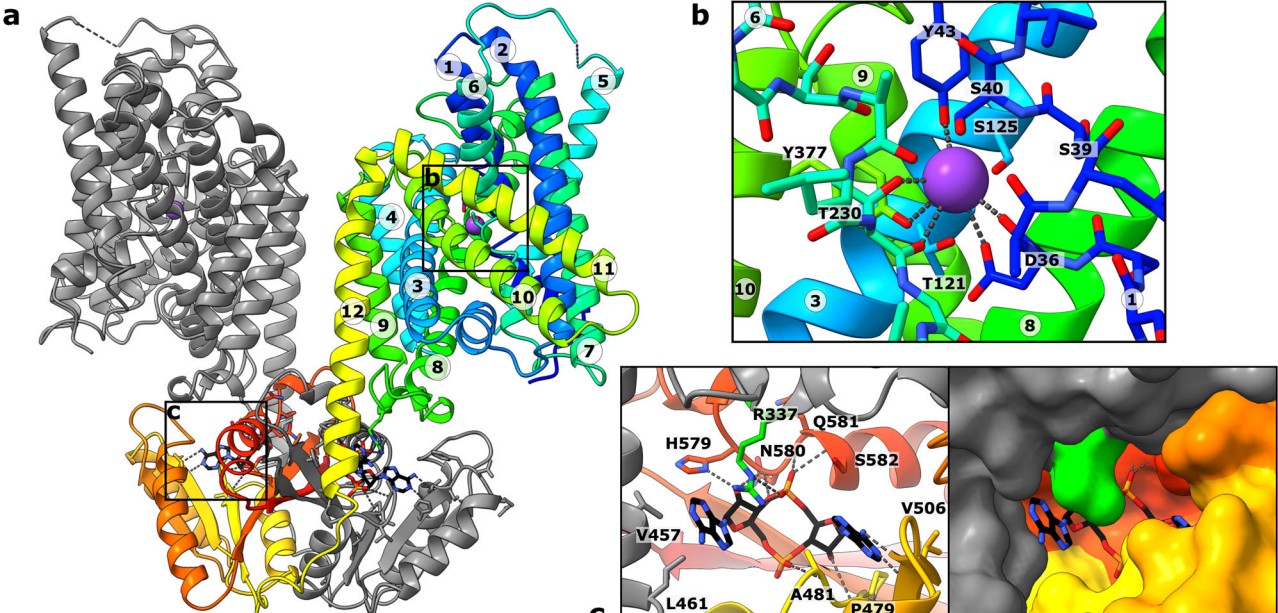

**Fig. 2 | Cryo-EM structure of KimA with c-di-AMP bound. a** KimA homodimer with c-di-AMP bound between the swapped CDs and TMDs of the grey and rainbow-coloured monomers. **b** K+ binding site of one TMD. The substrate potassium ion is coordinated by Tyr43, Asp36, Thr230 and Tyr377 with distances ranging between 2.8 and 3.3 Å. **c** The c-di-AMP binding pocket is mostly formed by residues of the cytosolic domain. Only Arg337 of the TMD strongly interacts with c-di-AMP. KimA was purified after co-production with CdaA and solubilized with DDM. A ten-fold molar excess of c-di-AMP was added after purification.

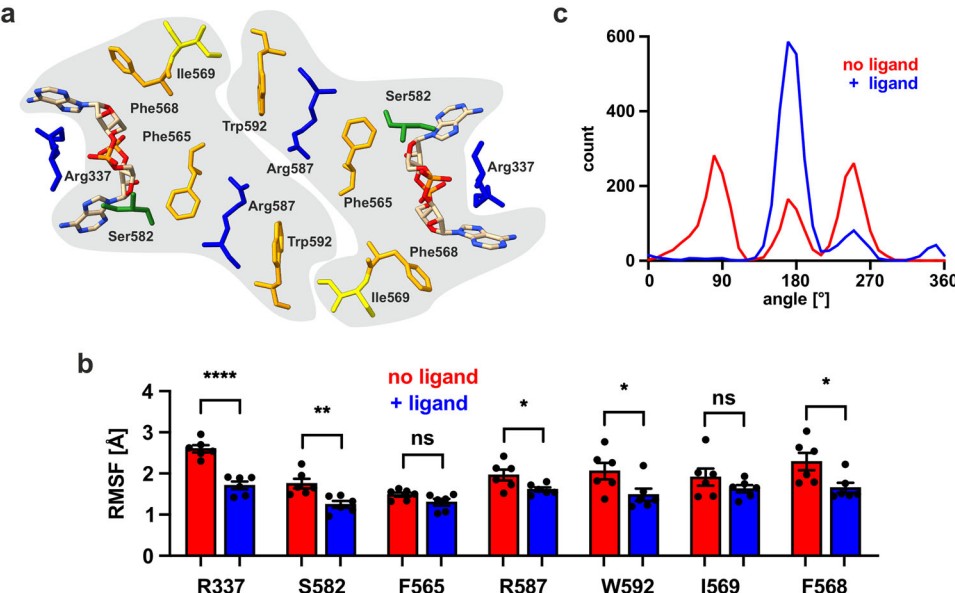

**Fig. 3 | Cooperativity network between cytosolic domains of KimA dimer. a** Top view of dimeric CDs highlighting bound c-di-AMP and residues linking both binding sites. Protomer affiliation highlighted in grey, basic residues blue, hydrophobic residues yellow, aromatic residues orange, polar residues green. **b** RMSF (Å) of residues connecting binding pockets in the absence (red) or presence (blue) of c-di-AMP. Data are from $3 \times 2.2$ μs simulations plotted as mean values ± SEM ($n = 6$). See Supplementary Fig. 3 for the raw values and the values from simulation with one KimA monomer bound to c-di-AMP and the other apo. For significance two-tailed t-test was performed with no adjustments (P values from left to right: <0.0001, 0.0052, 0.0853 (ns), 0.0334, 0.0457, 0.2363 (ns), 0.0265). **c** The presence of c-di-AMP has a considerable effect on the range of $X_2$ angles sampled by the Trp592 sidechain.

be the locking of KimA in the inward-occluded conformation by other means, namely the binding of c-di-AMP, which could prohibit K⁺ binding to the trans-inhibitory site from the cytosolic side. In fact, both cryo-EM maps of KimA in DDM and in amphipols clearly show two non-protein densities in the CDs, which were modelled as c-di-AMP (Supplementary Fig. 2b). Each binding site is formed mostly by the cytosolic domain of one monomer (Fig. 2c): One of the phosphodiesters of c-di-AMP is coordinated through two hydrogen bonds and a salt bridge between the non-esterified oxygen atoms and the amide groups of Gln581 and Ser582 and the guanidinium group of Arg337. One adenosine moiety forms hydrogen bonds with the backbone carbonyls of Val506 and Pro479. The hydroxyl group of the ribose of the second adenosine moiety forms hydrogen bonds with His579 and the backbone amide of Asn580. Only the second adenosine moiety is coordinated by residues of the neighbouring protomer; it is stacked between Val457 and Leu461 of TMH 12 that extends out of the membrane and Arg337, located in the loop between TMHs 8 and 9. Arg337 is the only residue of the TMD that engages in a strong interaction with c-di-AMP, making it a prime candidate for the transmission of the inhibitory effect of bound c-di-AMP.

### C-di-AMP binding to one binding site primes the other
The structural insights were used to further investigate the binding of c-di-AMP to KimA. In particular, we strove to understand how pre-binding of c-di-AMP facilitates further c-di-AMP binding, as suggested by DSF measurements. To this end, MD simulations were run of the KimA dimer either with or without c-di-AMP, with c-di-AMP remaining stably bound to KimA throughout the simulations where present, with an RMSD relative to the protein of 0.25 ± 0.07 nm over 3 repeats of ca. 2.2 μs each. Analysis of the data led to the identification of a cooperativity pathway between the two binding sites (Fig. 3a): Ser582, which is in direct contact with c-di-AMP, connects through Phe565 and Arg587 with Trp592. Trp592 links both CDs through a contact with Ile569 across the dimer interface. Ile569 is a direct neighbour of Phe568 which, like Phe565, delimits the c-di-AMP binding pocket. The root mean square fluctuation (RMSF) of almost all these residues was

significantly reduced compared to the apo-protein when c-di-AMP was present in both binding sites, suggesting a stabilising effect of c-di-AMP binding on the CD (Fig. 3b, Supplementary Movies 1 and 2). Surprisingly, if only one binding site was occupied by c-di-AMP, a similar stabilising effect was observed and all connecting residues of the apo-protomer also showed lower RMSF as if c-di-AMP was bound (Supplementary Fig. 3, grey highlighted, Supplementary Movie 3). Only the non-liganded Arg337 retains the high RMSF. It is possible that the more stable sidechain of Trp592 (Fig. 3c) restricts the motion of its counterpart Ile569 in the other protomer, which via the described network facilitates the priming of the second, empty c-di-AMP binding site.

### Arg337 increases the affinity for c-di-AMP but is not essential for the inhibition
As Arg337 was the only residue of the TMD identified to strongly interact with c-di-AMP in the binding pocket, it was suspected to transmit the inhibitory effect of c-di-AMP binding from the cytosolic domain to the TMD. DSF measurements showed that the mutated variant KimA_{R337A}, which should be unable to link c-di-AMP binding to the TMD, was still able to bind c-di-AMP when co-expressed with *cdaA*; however, a 20-fold excess was required for reaching the same thermal stabilisation as with a three-fold excess for the wild type (Fig. 4a), suggesting a decreased affinity for c-di-AMP.

To determine how the mutation affects the inhibition by c-di-AMP, in vivo potassium uptake studies were performed. In the presence of active CdaA, the potassium uptake velocity remained rather high (Fig. 4b), but the inhibition by c-di-AMP was not completely abolished. The $V_{max}$ value for K⁺ uptake by KimA_{R337A} in the presence of active CdaA was reduced by -44% to 111.7 ± 25.5 nmol*mg dw⁻¹*min⁻¹ when compared to the co-expression of KimA_{R337A} with inactive CdaA_{D171N}, where a $V_{max}$ value for K⁺ of 198.9 ± 35.8 nmol*mg dw⁻¹*min⁻¹ was determined (Table 1). To address whether the remaining inhibition was related to c-di-AMP binding to KimA or caused by the production of c-di-AMP in general, variant KimA_{A481WS582W} that should be unable to bind c-di-AMP because of a sterically blocked binding pocket, was

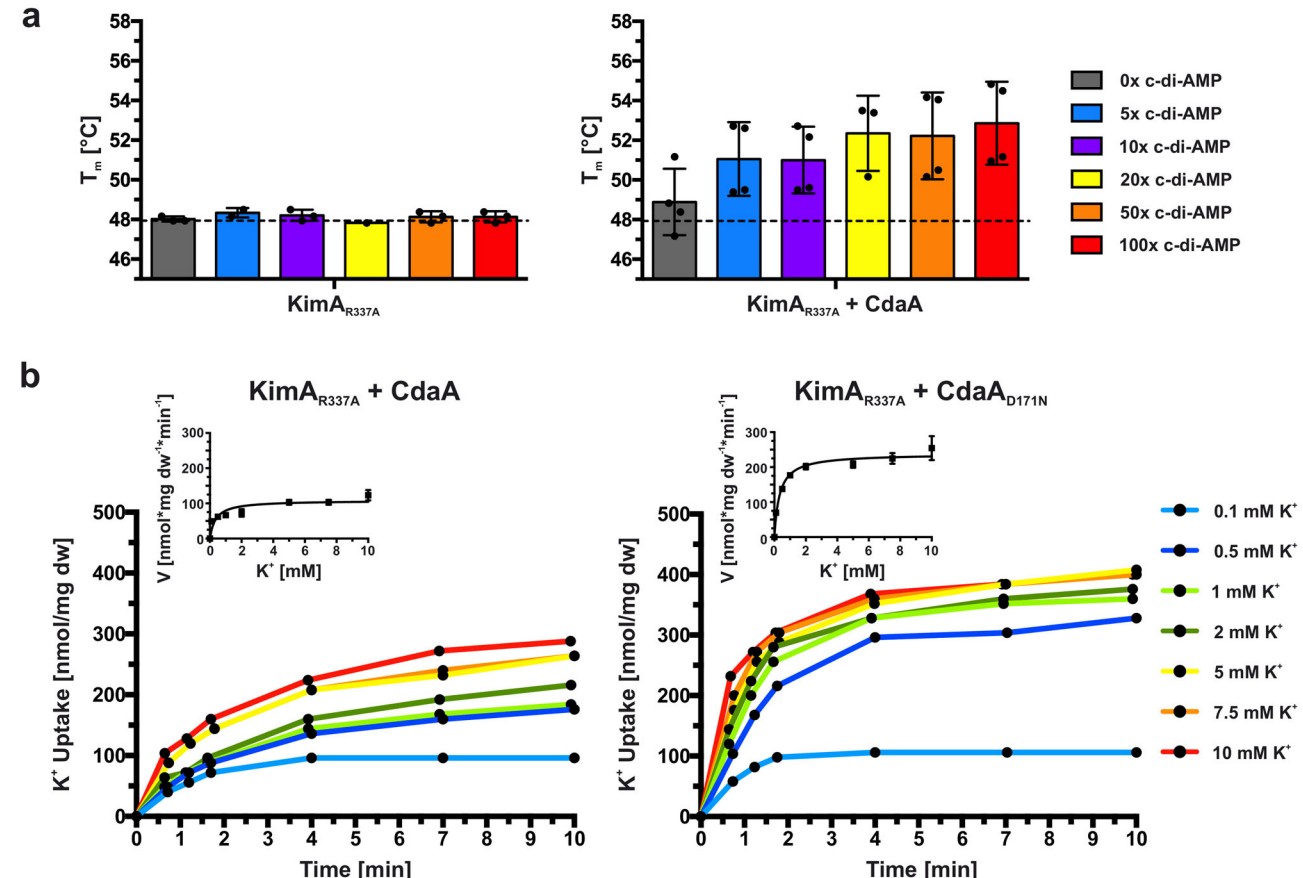

**Fig. 4 | Arg337 is essential for high-affinity binding of c-di-AMP. a** Melting temperatures of KimA R337A purified from cells with CdaA absent (KimA$_{R337A}$) or present (KimA$_{R337A}$ + CdaA), and incubated with an increasing c-di-AMP concentration given in x-fold molar excess over KimA. Determined with Differential Scanning Fluorometry (DSF). Dashed line indicates $T_m$ of KimA WT without c-di-AMP addition. Data point represents the average and error bars the standard deviation of measurements from at least biological triplicate (left: $n = 3$, right $n = 4$). **b** Whole-cell K$^+$ uptake assay in *E. coli* LB2003 cells producing KimA$_{R337A}$ with active (CdaA) or inactive (CdaA$_{D171N}$) diadenylate cyclase. Michaelis-Menten plot shown in graph with mean value ± SER. Representative experiment shown ($n = 3$).

analysed. The loss of c-di-AMP binding was confirmed by DSF measurements (Supplementary Fig. 4a). This loss of binding was accompanied by the insensitivity of the in vivo potassium uptake through KimA$_{A481WS582W}$ to inhibition by c-di-AMP (Supplementary Fig. 4b). The uptake velocities were comparable in the presence of an active and inactive cyclase, respectively (Table 1). In conclusion, Arg337 appears to be important for high-affinity c-di-AMP binding but is not necessarily required for communicating c-di-AMP binding to the TMD for inhibition.

## C-di-AMP-induced inhibition of KimA

C-di-AMP does not directly interact with the TMD, and long-range communication is required because the c-di-AMP binding site is ~37 Å away from the substrate K$^+$ binding site. The question remains how c-di-AMP binding to the CDs of KimA controls K$^+$ transport at a distance. To address this question, we further evaluated the MD simulations performed in the presence and absence of c-di-AMP. They showed a significant stabilisation of the whole intracellular loop between TMH8 and TMH9 upon c-di-AMP binding, not just Arg337 (Fig. 5a). This might suggest that inhibition by c-di-AMP is transmitted to the TMD via this loop.

The MD simulations further revealed that one consequence of c-di-AMP being removed from the system was a movement of TMH6 away from TMH3, leading to an opening of the TMD at the cytosolic end. This effect can be seen by measuring the distance between two residues sitting underneath the intracellular gate, Tyr118 (TMH3) and Asn237 (TMH6). These typically interacted closely (<4 Å) when c-di-

AMP is bound (Fig. 5b, Supplementary Fig. 5a), which likely contributes to locking the TMD in an inward-occluded conformation. When c-di-AMP was removed this interaction was broken (Fig. 5b), and the distance frequently shifted from about 4 Å to 8 Å (Supplementary Fig. 5b).

The increased distance of Tyr118 and Asn237 can be observed using principal component analysis (PCA) of the MD data (see Methods). This reveals that the principal changes observed in the MD data involve a rearrangement of the central TMHs, with TMH6 straightening and moving ca. 4 Å away from TMH3 and TMH8. This occurs in both eigenvector 1 (Fig. 5c) and eigenvector 2 (Supplementary Fig. 6). Analysis of the separate conditions reveals that these more open conformations are exclusively sampled by the no-liganded conditions, with the ligand-bound states instead resembling the input structure (Fig. 5c). Analysis of the MD data reveals that, whilst the initial bound K$^+$ remain tightly bound in the c-di-AMP-bound state, they are free to rapidly exchange with the bulk solvent in the c-di-AMP-free state, suggesting that the conformational changes seen in the PCA represent a switch to an inward-open conformation. In support of this observation, comparison of the inward-facing cavity reveals a clear widening in the ligand-free simulations (Supplementary Fig. 7a and b). This is accompanied by an increase in the number of waters present in the cavity (Supplementary Fig. 7c). Comparing the number of waters in the inward-facing cavity to the Tyr118-Asn237 distance reveals that the no-ligand system samples are considerably more open and more solvated states than the liganded system (Supplementary Fig. 7d). Comparison of the post-simulation ligand-bound and ligand-free states to the open-to-in structure of the structurally similar amino acid transporter BasC

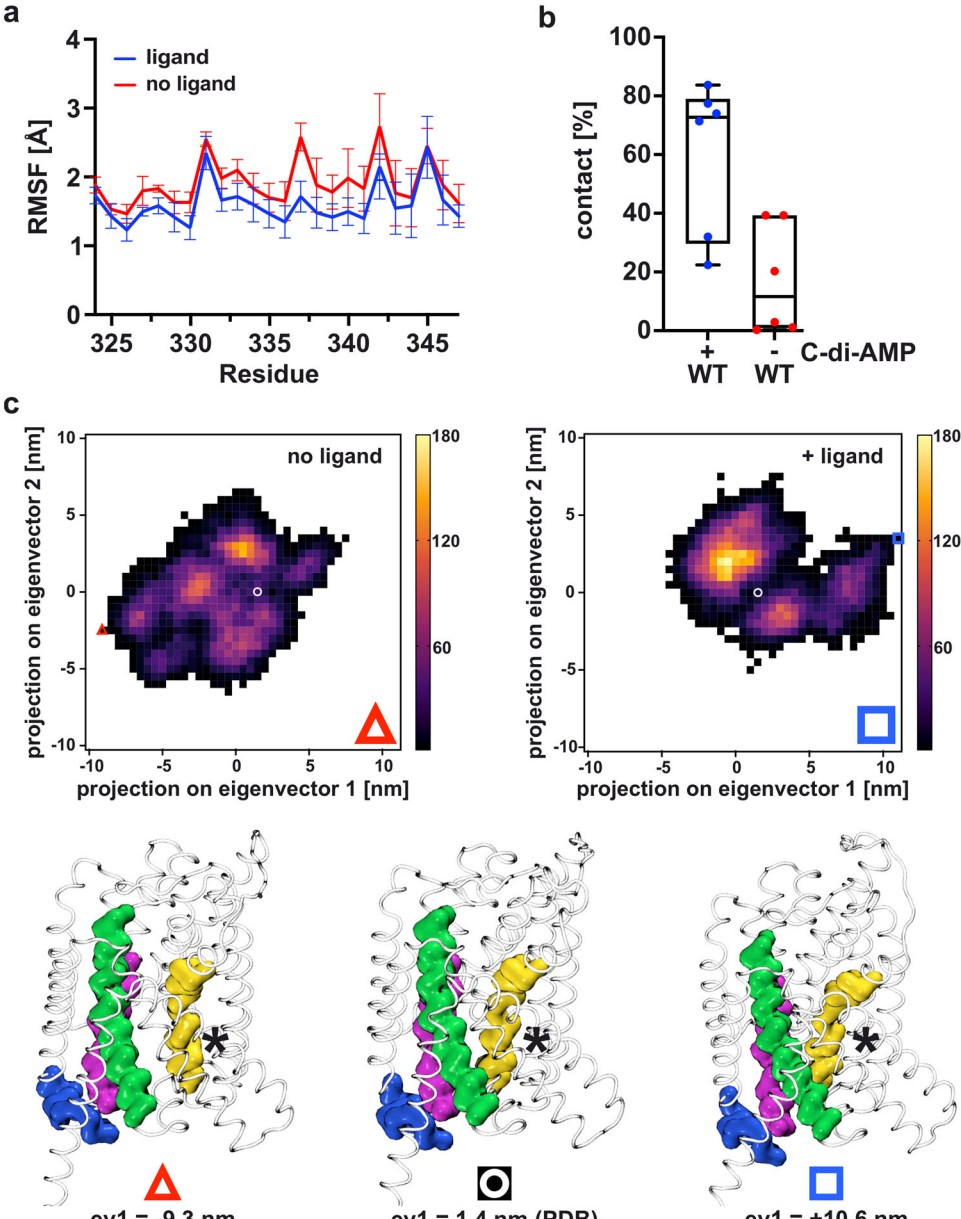

**Fig. 5 | Transduction of inhibition from CD to TMD upon c-di-AMP binding.**
**a** RMSF of binding loop between TMH 8 and 9 determined by MD simulations shown as mean value ± SEM ($n = 6$). **b** MD simulations reveal an increased likelihood of interaction (distance ≤4 Å) between Tyr118 and Asn237 upon c-di-AMP binding compared to when no ligand is bound. Interactions were analysed over $3 \times 2.2$ µs simulations for the system with and without c-di-AMP shown as box plot with minimum and maximum whiskers, median centre and upper/lower quartile box bounds ($n = 6$). **c** PCA analysis on the KimA C-alpha atoms. Analysis was run on the ligand and no-ligand data together, and then the sampling of eigenvectors 1 and 2 were plotted for each condition separately. Beneath the heatmaps are structures

representing the extreme states of eigenvector 1 (see red triangle and blue square on the heatmaps), as well as the input structure (white circle). With ligand bound, KimA is more likely to adopt conformations at higher eigenvector 1 and 2 values, close to the input conformation (white circle). Here, TMH6 (yellow) is angled, and closer to TMHs 3 and 8 (green and purple). When the ligand is absent, KimA samples additional states towards the bottom left of the heatmap, i.e. low eigenvector 1 and 2 values. Here, TMH6 has straightened and is further from TMH 3/8 (compare asterisks). Eigenvector 2 has a similar conformational change to eigenvector 1, see Supplementary Fig. 5.

(PDB 6F2W)[29], as identified using a FoldSeek[30] search on the no-ligand KimA structure, also supports the adoption of an inward open state for no-ligand bound KimA (Supplementary Fig. 8).

In agreement with the MD simulations, a mutation of Tyr118 or Asn237 to alanine led to a loss of inhibition by c-di-AMP in potassium uptake assays, while binding of c-di-AMP was still possible (Fig. 6a, Table 1, Supplementary Fig. 9). The mutations seem to have abolished the observed interaction from the MD simulations, and therefore reduce the ability of c-di-AMP to lock KimA in an inward-occluded state. The significantly increased $K_m$

of 19 mM and 8 mM for $KimA_{Y118A}$ and $KimA_{N237A}$, respectively, is in agreement with the assumption that these residues line the exit pathway to the cytosol and suggests their involvement in ion release.

In conclusion, the MD simulations together with the functional data suggest that the binding of c-di-AMP (Fig. 6b-1) communicates to the TMD via a stabilisation of the TMH8-TMH9 loop (Fig. 6b-2, blue), which in turn affects the positioning of TMH8 and TMH3 relative to TMH6 (Fig. 6b-3 purple, green, yellow). These conformations would appear to regulate the switching between inward-occluded and

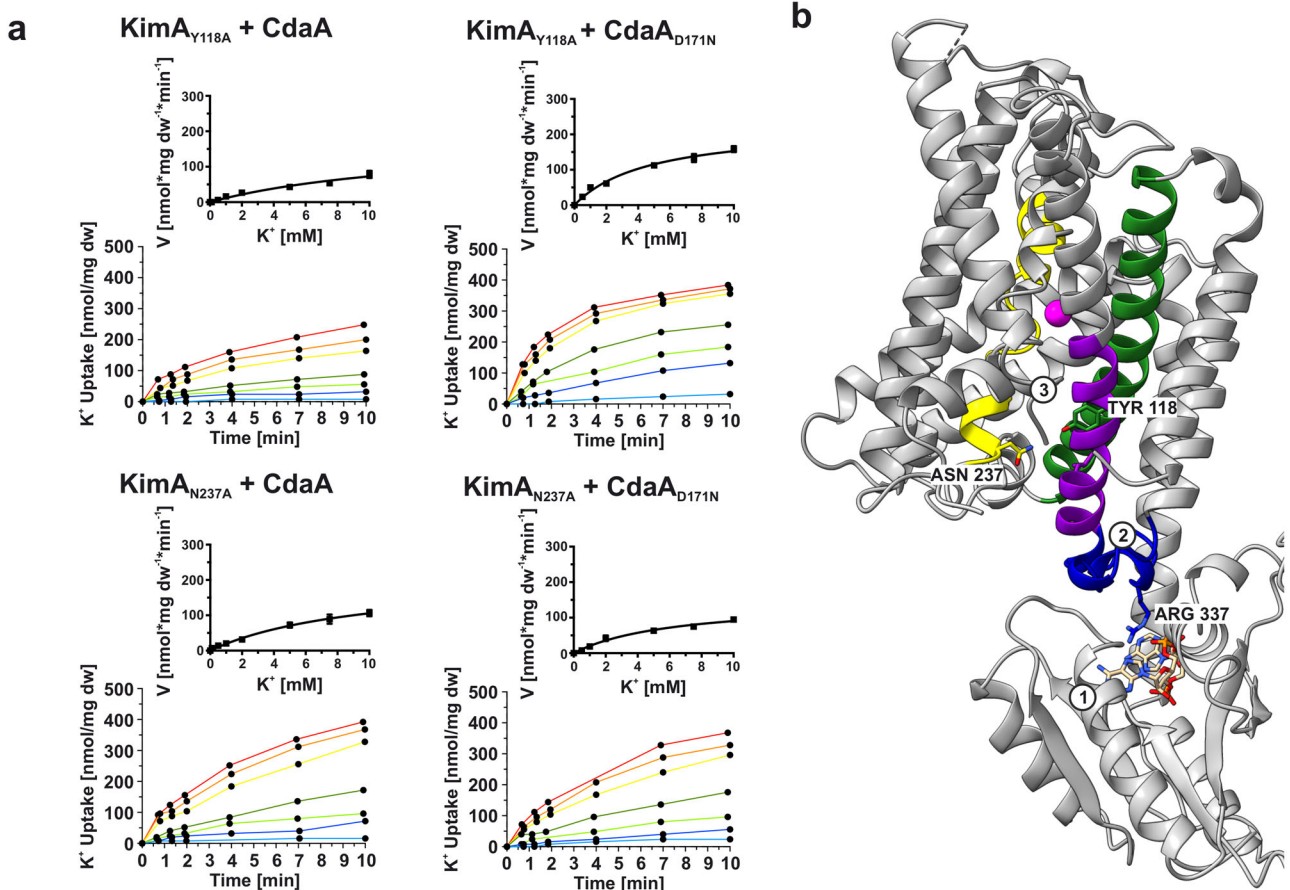

**Fig. 6 | Mutation of Tyr118 and Asn237 leads to reduced inhibition transduction in vivo. a** Reduced inhibition of potassium uptake in whole-cell K⁺ uptake assays when Tyr118 or Asn237 were mutated to alanine. K⁺ uptake for KimA_{Y118A} or KimA_{N237A} variant in the presence of active CdaA or inactive CdaA_{D171N}. Added K⁺: light blue: 0.1 mM, blue: 0.5 mM, light green: 1 mM, green: 2 mM, yellow: 5 mM, orange: 7.5 mM, red: 10 mM. Michaelis-Menten diagram shown in graph.

Representative experiment shown ($n = 3$). **b** Model for signal transduction: Upon c-di-AMP binding to CD (1), the binding loop (blue) rigidifies (2), TMH8 and 3 become restricted in their mobility, leading to more constant interaction between the cytosolic ends of TMH3 and 6, e.g. Tyr118 and Asn237 (3). The opening of the intracellular gate is abolished.

inward-open, thereby presenting a possible mechanism of c-di-AMP inhibition of KimA.

## Discussion

Our results provide first structural and functional insights into how c-di-AMP inhibits proteins of the KUP transporter family. Interestingly, the binding mode differs from other c-di-AMP-regulated transport proteins such as potassium channel KtrAB[19], osmolyte transporters OpuC[31] and OpuA[32], and potassium exporter KhtTU[22], where c-di-AMP binds at dimer interfaces in a symmetric binding pocket. In KimA, two c-di-AMP molecules bind to the dimer in an asymmetric binding pocket. Different to the binding sites in the RCK_C domains of KtrA and KhtT and in the CBS domains of OpuA and OpuC, in which two arginine residues, one per protomer, coordinate both phosphate esters, only one of the two phosphate esters of c-di-AMP is coordinated by an arginine (Arg337) in KimA. In fact, Arg337 clearly supports the high-affinity binding of c-di-AMP and the communication of its presence to the TMD, but is not essential for the ligand binding or its inhibitory function. Instead, our data suggest that the major linker to the TMD is the entire binding loop between TMH8 and 9, which as a whole becomes more restricted in its movement once c-di-AMP is bound. This stabilisation may limit the freedom of movement of TMH8, which then restrains TMH3 and TMH6, for instance through interactions between Leu325, Tyr118 and Asn237. MD simulations indicate that through these interactions, opening of the inner gate is hindered,

arresting KimA in an inward-occluded conformation. Consequently, the rocker switch movement can no longer take place (Fig. 7). When c-di-AMP is removed, KimA is once again able to open up to exchange K⁺ with the bulk solvent, as seen in our simulations. To corroborate the role of the binding loop, additional mutations (e.g., glycine) could be added to the loop in KimA_{R337A} to increase its flexibility. This could ultimately abolish inhibition by c-di-AMP despite its binding. However, this could also have unpredictable side effects on the stability of the protein, affecting the activity of KimA in unforeseen ways.

To elucidate how conserved the inhibition mechanism is among other KUPs we performed a structure-based sequence alignment with AlphaFold predictions of four other KUPs of gram-positive bacteria as well as Kup from the gram-negative *E. coli*, which have been functionally but not structurally characterized (Supplementary Fig. 10). While Kup from *E. coli* is not regulated by c-di-AMP because the second messenger is lacking in *E. coli*, KimA from *Listeria monocytogenes* (KimA^{Lmo})[26], and KupA and KupB from *Lactococcus lactis* IL1403[33] showed growth inhibition by c-di-AMP. Further, c-di-AMP binding was shown for KupA and with lower efficiency for KupB. The regulation of KimA from *Staphylococcus aureus* (KimA^{Sau}) by c-di-AMP remains unclear[26]. In agreement with the lack of c-di-AMP in *E. coli*, Kup from *E. coli* does not share any of the characteristic residues Arg337, Tyr118 or Asn237 with KimA from *B. subtilis*. In contrast, Arg337 is conserved in KimA^{Lmo}, KimA^{Sau} and KupA. KupB lacks an equivalent amino acid, which could explain the weaker c-di-AMP binding. KupB is still

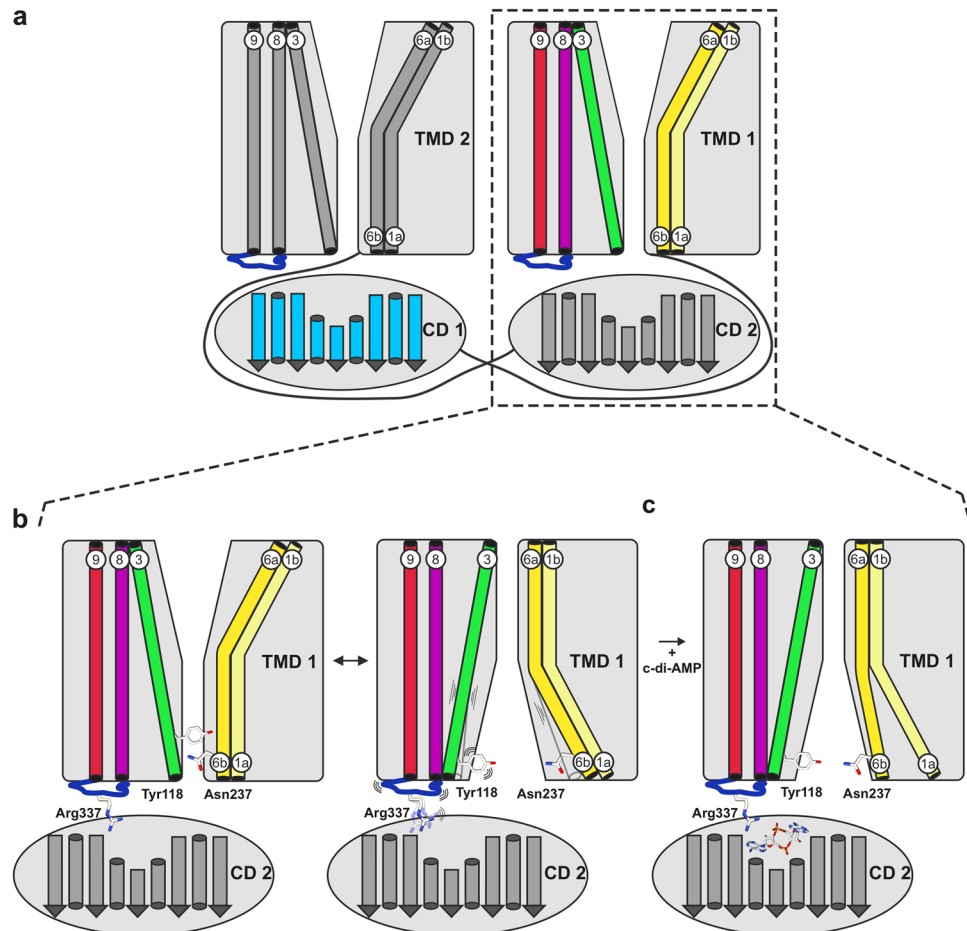

**Fig. 7 | Schematic model of c-di-AMP-induced inhibition of KimA. a** Dimer of KimA with swapped CDs. One protomer shown in grey, one with the decisive TMHs coloured as in Figs. 5 and 6. **b** KimA undergoes a rocker switch movement to fulfil proton-coupled potassium uptake. The binding loop is unrestricted in its movement. Tyr118 and Asn237 are only in loose contact. **c** C-di-AMP binding rigidifies the binding loop. Helix 8 leans on helix 3, stabilizing the interaction between helix 3 and 6 bringing Tyr118 and Asn237 closer. KimA is then locked in an inward-occluded conformation, preventing the opening of the inner gate.

inhibited by c-di-AMP, supporting our assumption that Arg337 is not required for transmission of the inhibition to the TMD. Residues Tyr118 and Asn237 are only conserved in KimA$^{Sau}$, while KupA, KupB and KimA$^{Lmo}$ share a glycine and a serine, respectively, at those positions. In the binding pocket, KimA$^{Sau}$ shows conservation for Pro479, His579, Asn580, Gln581 and similarity for Val457 and Ser582 (Ile/Thr in KimA$^{Sau}$, respectively). The KUPs show conservation only for Gln581 and like KimA$^{Sau}$ have a Thr at the position of Ser582, with the exception of Kup from *E. coli*, which has a lysine and leucine. Therefore, inhibition of KimA$^{Sau}$ by c-di-AMP is likely to be similar to KimA from *B. subtilis*, while for the other homologues some variations are to be expected.

A surprising observation was that c-di-AMP only bound to KimA *in vitro* when at least a proportion of c-di-AMP was co-purified along with KimA. Structurally, there is no obvious reason for this observation. The MD simulations showed that c-di-AMP stably binds to both the upright- and the tilted-dimer arrangement. The superposition of the previously solved structure in the absence of c-di-AMP with the c-di-AMP-bound structures does not show any significant conformational changes in the binding pocket, apart from a general stabilisation when c-di-AMP is bound. However, binding of the first c-di-AMP molecule by KimA appears to be unfavourable. A direct hand over of c-di-AMP from the cyclase could lower the energetic barrier for ligand binding. Binding of a second molecule then appears to be strongly cooperative within dimeric KimA. In agreement with this hypothesis, the only membrane-bound diadenylate cyclase CdaA in *B. subtilis* is described as the major cyclase maintaining the internal c-di-AMP concentration required for

cell growth[34]. In other bacteria like the pathogenic *L. monocytogenes*[35,36] and *S. aureus*[37] as well as *L. lactis*[38] CdaA is the only c-di-AMP cyclase. Since the majority of proteins regulated by c-di-AMP are membrane bound, the co-localisation of the cyclase appears advantageous if direct interaction is necessary.

In summary, we revealed key elements for the inhibition of potassium uptake through KimA by c-di-AMP. We show the structure of a KUP transporter with bound secondary messenger c-di-AMP and confirmed its inhibitory binding. Based on our findings we propose a network for the inhibition of KimA by c-di-AMP at a distance. Structures of outward-open and inward-open KimA are necessary to confirm the proposed transport cycle and validate the mode of inhibition by c-di-AMP.

## Methods
### Cloning of KimA variants using site-directed mutagenesis
Point mutations were introduced into pB24KimA using site-directed mutagenesis (SDM). Primer pairs of 20–40 bp length including a 1–3 bp mismatch were used to introduce point mutations (Supplementary Table 1). PCR product was digested with DpnI to remove template DNA and subsequently used to transform *E. coli* DH5α. Colonies grown on LB agar plates with 100 μg/ml ampicillin were picked and cultured in 5 ml LB with 100 μg/ml ampicillin over day. Plasmids were isolated from cells using NucleoSpin Plasmid, Mini kit for plasmid DNA (Macherey-Nagel). Correct mutation of the plasmids was verified by sequencing performed by MicroSynth GmbH Göttingen.

## Expression and protein purification of KimA

A colony of *E. coli* LB2003 cells transformed with the expression plasmids for KimA or variants thereof (pB24KimA) and, if indicated, CdaA variants (pB33CdaA) where grown at 37 °C o/d with 180 rpm shaking in 5 ml KML (1% KCl, 1% tryptone, 0.5% yeast extract (w/v)) with 100 μg/ml ampicillin and 30 μg/ml chloramphenicol if needed. An overnight culture of 200 ml KML with respective antibiotics was inoculated from the o/d culture and incubated at 37 °C with 180 rpm. 6 l KML were inoculated with the o/n culture to an $OD_{600}$ of 0.1 and incubated at 37 °C with 180 rpm. Gene expression was induced with 0.002% arabinose at an $OD_{600}$ of 1. Cells were grown for 1.5 h and then harvested by centrifugation. Cells were resuspended in buffer containing 420 mM NaCl, 180 mM KCl, 50 mM Tris-HCl pH 8 supplemented with 1 mM EDTA, 0.1 mM PMSF, 0.3 mM benzamidine and DNase I. Cells were disrupted by passing through a cell homogeniser (Stansted Pressure Cell Homogeniser FPG 12800) at 1 kbar. Cell debris was removed by centrifuging at $15,000 \times g$ for 15 min. Membranes were harvested o/n by centrifuging at $100,000 \times g$. Membranes were resuspended in aforementioned buffer (100 mg/ml) and homogenised. Membranes were solubilised for 1 h at 4 °C with 1% of DDM (Glycon). Unsolubilised proteins were removed by centrifuging 30 min at $135,000 \times g$. Supernatant was incubated with 2 ml $Ni^{2+}$-NTA resin for 1 h at 4 °C. Ni-NTA was washed with 50 column volumes of buffer containing 140 mM NaCl, 60 mM KCl, 20 mM Tris-HCl pH 8 and 0.04% DDM supplemented with 50 mM imidazole. Protein was eluted by using 500 mM imidazole in aforementioned buffer. Protein was further purified via size exclusion chromatography using a Superose6 Increase column (GE Healthcare/Cytiva) preequilibrated with aforementioned buffer without imidazole addition. Protein was concentrated and incubated with c-di-AMP if needed for DSF, HPLC-MS and cryo-EM experiments.

## Preparation of KimA in amphipols

Protein expression and purification were performed as described above until binding to $Ni^{2+}$-NTA. The beads were washed with 50 column volumes of buffer (140 mM NaCl, 60 mM KCl, 20 mM Tris-HCl, pH 8) containing 60 mM imidazole and a reduced DDM concentration of 0.025%. KimA was eluted by making use of the C-terminal HRV-3C-Protease cleavage site. Therefore, the beads were incubated with 3C-Protease for 1.5 h at 4 °C. The beads were washed twice with buffer containing 0.025% DDM. The elution fraction containing KimA was concentrated and filtered. For the detergent/amphipol exchange, KimA at a concentration of 12 mg/ml and amphipols PMAL C8 (10% in water; Anatrace) were incubated at w/w ratio of 1:10 for 1 h at 4 °C. To remove excess detergent, the sample was incubated with biobeads at a detergent-to-biobeads weight ratio of 1:100 overnight. The biobeads were removed and the sample was loaded onto a Superose6 Increase 200 10/300 GL column (GE Healthcare) equilibrated to cryo-EM buffer (50 mM NaCl, 50 mM KCl, 20 mM Tris-HCl, pH 8). KimA-containing fractions were pooled and concentrated to 2.25 mg/ml for cryo-EM.

## In vivo whole-cell potassium uptake

Protocol was adapted from[25,39]. A colony of *E. coli* LB2003 cells transformed with the expression plasmids for KimA variants (pB24KimA) and CdaA variants (pB33CdaA) was grown at 37 °C o/d with 180 rpm shaking in 5 ml KML with 100 μg/ml ampicillin and 30 μg/ml chloramphenicol. 75 ml of $K_{30}$ minimal media supplemented with the respective antibiotics were inoculated with 3 ml of o/d KML culture and incubated o/n at 37 °C. 500 ml of $K_{30}$ minimal media with antibiotics were inoculated to an $OD_{600}$ of 0.15 and incubated at 37 °C with 180 rpm. Gene expression was induced at an $OD_{600}$ of 0.4–0.6 with 0.002% arabinose. Cells were grown 1.5 h before being harvested with $6000 \times g$ at 20 °C for 10 min. The cell pellet was washed twice in 10 ml 120 mM Tris-HCl pH 8. Cells were adjusted to an $OD_{600}$ of 30 and incubated for 5 min at 37 °C in a water bath (130 rpm shaking). To

permeabilize cells for internal potassium, 1 mM EDTA was added and cells were incubated at 37 °C for 7 min shaking at 130 rpm in a water bath. Cells were centrifuged with $4500 \times g$ for 7 min at 20 °C and washed twice in 200 mM HEPES TEA pH 7.5 to remove EDTA and internal potassium. Cells were adjusted to an $OD_{600}$ of 30. For uptake, cells were diluted to an $OD_{600}$ of 3 with 200 mM HEPES TEA pH 7.5 and energized with 0.2% glycerol and 0.002% arabinose. Different potassium concentrations (0.1, 0.5, 1, 2, 5, 7.5, 10, 15 mM) were added and 1 ml samples were taken at 0, 1, 2, 4, 7, 10 min after addition. Potassium uptake was stopped as cells were centrifuged through 200 μl of silicone oil ($\rho = 1.04$) at $17,000 \times g$ for 2 min. Media and oil was removed and the 1.5 ml reaction tube tip containing the cell pellet was cut off. Cell pellet was resuspended in 1 ml of 5% TCA. Cells were lysed by freezing at −20 °C and subsequent boiling at 95 °C for 10 min. Supernatant was diluted with 3 ml of 6.7 mM CsCl and 4 ml 5 mM CsCl. Cell debris was pelleted by centrifuging 20 min at $4000 \times g$. Internal potassium concentration was determined using flame atomic absorption spectroscopy (F-AAS). Graphs and kinetic analysis were made with Prism 6.0e GraphPad.

## Differential Scanning Fluorometry/Thermal Shift Assay (DSF/TSA)

When ligands bind their target, they stabilize or destabilize the protein leading to a shifted protein melting temperature. The melting temperature is detected via the increasing fluorescence of the 7-diethylamino-3-(4-maleimidophenyl)-4-methylcoumarin (CPM) dye which binds cysteine residues that become accessible during the unfolding process[40,41]. 0.5 mg/ml purified KimA protein, co-expressed with and without *cdaA*, was incubated with 0–100x molar excess of c-di-AMP overnight at 4 °C. The next day 16.67 μg/ml CPM dye was added with subsequent incubation for 15 min on ice in the dark. 10 min centrifugation at $17,000 \times g$ at 4 °C removed aggregates. Melting curves were recorded in a Rotor-Gene Q 5Plex HRM system (Qiagen) using a temperature range of 25–85 °C with 1 °C steps each 30 s and 90 s prewarm. CPM was excited at 365 nm and emission was detected at 460 nm. Samples were measured in technical triplicates. Melting temperatures were determined at the inflection point of the resulting fluorescence curve using Prism 6.0e GraphPad. Graphs were plotted with Prism 6.0e GraphPad.

## Cryo-EM specimen preparation and data acquisition

UltrAuFoil R1.2/1.3 400 mesh gold grids were glow-discharged twice before the application of 3 μl of either a 4.0 mg/ml solution of KimA in DDM or 2.25 mg/ml KimA reconstituted in amphipols. The sample was then plunge-frozen using a FEI Vitrobot Mark IV. The chamber and Whatman 595 blotting paper were equilibrated at 4 °C and 100% relative humidity.

A Titan Krios (Thermo Scientific) equipped with a Gatan K3 camera in counting mode and energy filter was used for imaging with the software EPU 2.9 (Thermo Scientific). The fluence over a broken hole was adjusted to 1.1 electrons/$Å^2$ per frame. Micrographs were acquired as 50-frame movie stacks in 2.7 s or 2.3 s exposures, respectively, at a nominal magnification of ×105,000 with a resulting pixel size of 0.83 Å. Defocus values were set in the range of −1.1 to −2.5 μm.

## Image processing and model refinement

Micrographs were processed using Relion-3.1[42] (Supplementary Fig. 11). The Relion implementation of MotionCor2[43] was used for drift correction and dose weighting. Gctf[44] was used for the initial CTF estimation. Initially, 950k coordinates were picked from 2349 micrographs by Topaz[45] after training the neural network with 726 particles that were manually picked from 30 micrographs of KimA in DDM. After two rounds of 3D classification 332k particles remained that yielded a 3.6 Å map of KimA. Per particle drift correction and dose-weighting during Bayesian polishing[46] and two iterations of CTF refinement

improved the resolution to 3.5 Å. The two protomers in the dimer did not show deviations from C2 symmetry and applying the symmetry yielded a final map at 3.3 Å resolution (Supplementary Fig. 12a).

In the case of KimA in amphipols, 5.18 million coordinates were picked from 4303 micrographs by Topaz after training the neural network with 530 particles that were manually selected from 20 micrographs. Bad picks were removed by two consecutive 3D classifications yielding 1.99 million and 861k particles, respectively. A lowpass filtered map of KimA solubilized in SMA[25] was used as an initial 3D template. Additional rounds of 3D classification resulted in a set of 296k particles from the 3417 best micrographs yielding a 4.1 Å map of KimA. The resolution was also improved by Bayesian polishing and two iterations of CTF refinement to 4.0 Å (C1) and 3.8 Å (C2 symmetry) (Supplementary Fig. 12b). Further improvement of the map was achieved through focused classification of symmetry expanded particles. The particles of the last refinement were C2 symmetry expanded. Then density outside a wide mask that contained one TMD and the cytosolic domain of the other protomer was subtracted from the particle image with a Relion Particle Subtraction job. 3D classification of these particles with four classes, local searches only and the same mask used for the particle subtraction as a reference mask yielded one class with 307k particles. 3D-autorefinement of these particles resulted in a 3.7 Å map of one half of the dimer (Supplementary Fig. 12c).

Individual domains of the model of KimA in SMA (pdb ID 6s3k)[25] were rigid body fitted into the map of the upright dimer with *Coot*[47] and manually rebuilt. Models were optimised using Phenix real space refinement[48] (Supplementary Table 2).

## Molecular dynamics simulations

Atomistic simulations were built using the coordinates of dimeric KimA bound to c-di-AMP from this study. The systems were described with the CHARMM36m force field[49,50] and built into 6:3:1 POPE, POPG, cardiolipin membranes with TIP3P waters and $K^+$ and $Cl^-$ to 150 mM, using CHARMM-GUI[51,52]. Three bound $K^+$ from each subunit were preserved from the input structure. The c-di-AMP molecules were parameterised in CHARMM-GUI, and either included in both subunits, a single subunit, or in neither subunit. Each system was minimized and equilibrated according to the standard CHARMM-GUI protocol. For the system with no c-di-AMP molecule present, the molecule was deleted and an additional 30 ns equilibration simulation was run with restraints on the protein backbone. Production simulations were run in the NPT ensemble, with temperatures held at 303.5 K using a velocity-rescale thermostat and a coupling constant of 1 ps, and pressure maintained at 1 bar using a semi-isotropic Parrinello-Rahman pressure coupling with a coupling constant of 5 ps[53,54]. Short range van der Waals and electrostatics were cut-off at 1.2 nm. Simulations were run in triplicate, to ca. 2.2 μs for the wild-type systems with c-di-AMP present or apo, or to 500 ns for asymmetric c-di-AMP occupancy systems. In total, ca. 14.7 μs of data were gathered.

PCA was carried out on the C-alpha atoms of each KimA monomer. The trajectories for each subunit and each of the 3 repeats were concatenated before analysis for a total of 13.2 μs sampling. A covariance matrix using all data was constructed using gmx covar, and eigenvectors were analysed using gmx anaeig. Ca. 1700 eigenvectors were found, of which eigenvectors 1 and 2 contributed ca. 45% of the total variance (see Supplementary Fig. 13). Using the data for each of the ligand and no ligand conditions, separate 2D heatmaps of eigenvector 1 and 2 sampling were constructed using Prism.

All simulations were run in Gromacs 2020.1[55]. Data were analysed using Gromacs tools and VMD[56]. Plots were made using Prism 9 (GraphPad). Where used, homology models were built using SwissModel[57].

## Reporting summary

Further information on research design is available in the Nature Portfolio Reporting Summary linked to this article.

## Data availability

Data supporting the findings of this manuscript are available from the corresponding authors upon request. The cryo-EM maps were deposited in the Electron Microscopy Data Bank (EMDB) under accession codes EMD-15895 (KimA with c-di-AMP in amphipols) and EMDB-15894 (KimA with c-di-AMP in DDM). Atomic coordinates have been deposited in the Protein Data Bank (PDB) under accession codes 8B71 (KimA with c-di-AMP in amphipols) and 8B70 (KimA with c-di-AMP in DDM). 6S3K (KimA in SMA) used for model building is available in the PDB. Source data are provided as a Source data file. Source data are provided with this paper.

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

## Acknowledgements

We thank the Central Electron Microscopy Facility of the Max Planck Institute of Biophysics for cryo-EM infrastructure and technical support, and Juan Castillo-Hernández and Özkan Yildiz for support in cryo-EM data processing. This work was supported by the German Research Foundation via SPP1879 to J.V. and I.H. (VO 1449/1-1 and HA 6322/4-1), and the Heisenberg programme to I.H. (HA 6322/5-1). Research in PJS's lab is also funded by Wellcome (208361/Z/17/Z) and BBSRC (BB/ P01948X/1, BB/R002517/1 and BB/S003339/1). This project made use of time on ARCHER2 and JADE2 granted via the UK High-End Computing Consortium for Biomolecular Simulation, HECBioSim (http://hecbiosim. ac.uk), supported by EPSRC (grant no. EP/R029407/1). This project also used Athena and Sulis at HPC Midlands+, which were funded by the EPSRC on grants EP/P020232/1 and EP/T022108/1. We thank the

University of Warwick Scientific Computing Research Technology Platform for computational access.

## Author contributions

J.V. and I.H. conceived this study. All authors designed the experiments. M.F.F. performed in vivo and DSF experiments. M.F.F., Y.H. and I.T. purified KimA for cryo-EM. J.P.W, J.S.S. and J.V. performed the cryo-EM analysis, and built and validated the atomic models. R.A.C. performed MD simulations. All authors participated in the data analysis. M.F.F., J.P.W. and Y.H. wrote the initial draft, all authors participated in manuscript editing and revision. P.J.S., J.V. and I.H. supervised work and secured the funding for this work.

## Funding

## Competing interests

The authors declare no competing interests.
