## [Peer Review File · Nature Communications]

Cyclic di-AMP traps proton-coupled K⁺ transporters of the KUP family in an inward-occluded conformationReviewers' Comments:

Reviewer #1:

Remarks to the Author:

The authors have determined the structure of the potassium transporter KimA in complex with the second messenger cyclic di-AMP. Using an excellent combination of structural biology, computer modeling, and biochemical analysis they propose a model for the mechanism of KimA inhibition by c-di-AMP.

The study will be highly relevant to the field of second messenger signaling which is very dynamic.

I have a few comments:

l.70: inhibition of KimA by c-di-AMP, the references seem to be wrong, ref. 19 is about Ktr and was published before KimA was even discovered. It should be ref. 20 here!

refs. 30, 31, 40: Please add article/page numbers!

Reviewer #2:

Remarks to the Author:

Fuss et al. determined the binding site of c-di-AMP in KimA K transporters and overall structure of purified KimA using cryo-EM. The authors overcome the problem on the unbinding of c-di-AMP to KimA by coproducing it with CdaA in E. coli LB2003 mutant. Several crucial residues participate in the conformational changes and changes of binding affinity to c-di-GMP. The replacement of Tyr118 or Asn237 by alanine gave a loss of inhibition by c-di-AMP. They may involve in the ability of c-di-AMP to lock KimA in an inward-occluded state. Based on the MD simulation, they make the switching conformation model between inward-occluded and inward-open. E. coli Kup is not regulated by c-di-AMP, and therefore the Kup has no corresponding residues Arg337, Tyr118 or Asn237 with Bacillus subtilis KimA. These indicate that the regulation of Kup-type K transporter may happen in B. subtilis. This study provides knowledge useful for future research as knowledge of the structure and function of the K transporter KUP.

Minor Comments

L74 "adenylate cyclase variant" should be "diadenylate cyclase variant".

L370 "its binding in vivo and in vitro." Where are the "in vivo" results shown?

Reviewer #3:

Remarks to the Author:

In their manuscript, Fuss et al. combine biochemical measurements of potassium transport, cryoEM and MD simulations to study the inhibition of K⁺ transport of the KimA symporter by cyclic di-AMP. The main finding is that the presence of c-di-AMP in the binding pocket(s) of KimA leads to stabilization of the inward-occluded conformation, which would slow down the K⁺ transport. In contrast, when c-di-AMP is removed, MD simulations show a spontaneous transition of the transporter to the outward-open conformation and unbinding of the K⁺ ions. This is corroborated by mutational studies of Asn237 and Tyr118 residues that act as gates in the inward-occluded conformation. Additionally, the cooperativity in c-di-AMP binding is suggested, based on MD simulations and experiments.

Overall, the manuscript presents an interesting story, and the proposed mechanism of inhibition of KimA by c-di-AMP is plausible and supported by the data. I have however a number of points that should be resolved before the publication of the manuscript. I will focus mostly on MD simulations

1. The investigation of the 'cooperativity' or 'priming' of one binding site by c-di-AMP binding to the other is not very convincing. The authors say that 'RMSF of all these residues were significantly reduced compared to the apo-protein..' based on the table given in Fig 3 b. This table includes only averaged RMSF values from simulations with some (which?) error estimate. Closer inspection reveals that some of these values overlap within their error bars (1st row vs 2nd/3rd): Ile569, Phe565 and Phe568 in multiple instances do not seem to differ (within the error estimates provided). If the authors wanna say something about significance, I suggest showing individual data points from each simulations, and calculating p values for each compared pair. Visually, that would be better seen on a scatter plot, rather than in a table with many values. Further, it is not clear how the simulations without the ligand (apo state) were initiated: was c-di-AMP deleted from its binding site or did the authors used they previously published structure (ref 25?). A removal of a ligand from an existing structure would naturally cause large fluctuations of residues involving in its binding; it would be therefore of interest to compare the RMSF values with those from previous simulations of a 'true' apo conformation.

2. The observation that KimA spontaneously transitions toward the inward-open conformation after removing (deleting?) c-di-AMP is very interesting, and deserves some additional analysis in my opinion. For example, the size of the intracellular opening, its water accessibility and potassium unbinding pathway(s) should be analyzed. It would be also very interesting to put this transition into a broader perspective - since KimA has the popular LeuT fold, I'd assume that structures of related transporters in the inward-open conformation might exist. A comparison of such structures would be therefore very interesting and could deepen our understanding not only of KimA, but also of other proteins. Further, in a similar spirit to the previous comment, can the authors clarify why this transitions occurs in current simulations, but did not seem to occur in previous MD simulations (ref 25), although both proteins are in the apo conformation?

3. The PCA analysis could be improved. It is not clear whether the PCA was done on all trajectories together, or those with and without ligand separately. If the latter, then the meaning of eigenvectors 1 is different between these two sets (it is a different vector). It is also not clear what is the color code in the Supplementary Figure 10. I'd recommend performing PCA on all trajectories concatenated together, which should show clear basins corresponding to the 'inward-open' and 'inward-occluded' states.

4. The behavior of the used mutant R337A seems to be complicated: based on biochemical data, the authors conclude that R337 is important for high affinity binding of c-di-AMP, but its not necessary required for communicating c-di-AMP binding to the TMD, because the mutation reduced K⁺ transport (induced by c-di-AMP) by 44%. However, MD simulations suggest that, in terms of loop dynamics, R337A behaves similarly to the ligand-bound protein (Fig 5 a). Therefore, is it possible that R337 is (also) involved in the communication between c-di-AMP binding site and TMD, but the alanine mutation mimics (to some extend) R337 reduced dynamics upon c-di-AMP binding? Do R337A MD simulations also show the transition to the inward-open conformation?

5. If the reduced flexibility of the loop is the underlying reason for c-di-AMP mediated inhibition of KimA, increasing its flexibility, e.g. via glycine mutations, should abolish the effect of c-di-AMP. Can the author at least discuss it (ideally perform such experiments, but I won't push for it).

Reviewer #4:

Remarks to the Author:

The authors have presented a multidisciplinary study on the effect of c-di-AMP on the structural conformation and dynamics of a KUP transporter, KimA, using a combination of cryoEM, molecular dynamics, and functional assays. Overall, the data supports the authors interpretation and I have no

major concerns. The two cryoEM structures presented are of good quality and the secondary structure appropriately modelled. The ligand density has also allowed the authors to place c-di-AMP into a suitable pose. However, there are a few areas in which the manuscript and structures could potentially be improved, listed below are some minor concerns and suggestions. Issues in modeling below should be corrected in order to uphold the rigor and quality expected in the field, but I note that they have not affected the interpretations made by the authors and therefore I would categorize them as minor revisions with regards to reviewing the manuscript as a whole.

-In PDB:8B70/EMD15894 (DDM map), residue R462, near the c-di-AMP binding site, is modeled outside the EM density and should be remodeled into the continuous density extending towards c-di-AMP.

-Although the use of C2 symmetry in the final 3D refinement may result in a higher calculated global resolution, it may be concealing subtle asymmetric differences between the monomers. This is particularly salient for the present study as the authors note a level of cooperativity in c-di-AMP binding. The authors should carefully review their C1 generated maps to be sure they indeed have symmetry before forcing C2 symmetry during processing. It may be worth commenting within the manuscript that C1 displays no asymmetry, or if they find that it does then elaborate on their findings.

-There are several residues (e.g., K466, E511, K402, N244, E521, K402, R573 in PDB:8B71, and K515, D509, N244 in PDB:8B70) on the periphery that do not have sufficient density to justify modeling the sidechains. The authors should review their structures and stub side chains outside of the EM density.

-Please make note of the residues shown in the supplemental videos (V1-V3) by either directly labeling within the video, identifying by color in the Supplementary Materials legend, or by remaking the video to be in a similar vantage point as figure 3a and noting in the V1-3 legends that the region is viewed as in figure 3a.

-Add a panel to the supplement to show the cryoEM map-to-model fit of c-di-AMP and surrounding residues, as well as for the K⁺ site highlighted in figure 2.

-Two of the modeled DDM in PDB:8B70 at the dimer interface (B/903, A/1003) do not have EM density to support their modeling, their inclusion is unnecessary, and they should be removed.

-The summary schematic of figure 6 is a bit confusing. I suspect the authors were attempting a schematic similar to the one shown in the SMALP paper (ref 25), but the disjunction of the TM and CD domains makes the schematic unclear, especially in absence of labels of the structural elements/domains.

-In supplementary fig. 3 the authors may want to consider adding the moving average of the traces to the graphs.

Reviewer #1 (Remarks to the Author):

The authors have determined the structure of the potassium transporter KimA in complex with the second messenger cyclic di-AMP. Using an excellent combination of structural biology, computer modeling, and biochemical analysis they propose a model for the mechanism of KimA inhibition by c-di-AMP.

The study will be highly relevant to the field of second messenger signaling which is very dynamic.

I have a few comments:

l.70: inhibition of KimA by c-di-AMP, the references seem to be wrong, ref. 19 is about Ktr and was published before KimA was even discovered. It should be ref. 20 here!

Thank you. Corrected.

refs. 30, 31, 40: Please add article/page numbers!

Thanks, the article numbers were included.

Reviewer #2 (Remarks to the Author):

Fuss et al. determined the binding site of c-di-AMP in KimA K transporters and overall structure of purified KimA using cryo-EM. The authors overcome the problem on the unbinding of c-di-AMP to KimA by coproducing it with CdaA in E. coli LB2003 mutant. Several crucial residues participate in the conformational changes and changes of binding affinity to c-di-GMP. The replacement of Tyr118 or Asn237 by alanine gave a loss of inhibition by c-di-AMP. They may involve in the ability of c-di-AMP to lock KimA in an inward-occluded state. Based on the MD simulation, they make the switching conformation model between inward-occluded and inward-open. E. coli Kup is not regulated by c-di-AMP, and therefore the Kup has no corresponding residues Arg337, Tyr118 or Asn237 with Bacillus subtilis KimA. These indicate that the regulation of Kup-type K transporter may happen in B. subtilis. This study provides knowledge useful for future research as knowledge of the structure and function of the K transporter KUP.

Minor Comments

L74 “adenylate cyclase variant” should be “diadenylate cyclase variant”.

Corrected.

L370 “its binding in vivo and in vitro.” Where are the “in vivo” results shown?

Thanks for asking. We intended to say that the inhibition by binding has been shown. This has been corrected to read “We show the first structure of a KUP transporter with bound secondary messenger c-di-AMP and confirmed its inhibitory binding.”

Reviewer #3 (Remarks to the Author):

In their manuscript, Fuss et al. combine biochemical measurements of potassium transport, cryoEM and MD simulations to study the inhibition of K⁺ transport of the KimA symporter by cyclic di-AMP. The main finding is that the presence of c-di-AMP in the binding pocket(s) of KimA leads to stabilization of the inward-occluded conformation, which would slow down the K⁺ transport. In contrast, when c-di-AMP is removed, MD simulations show a spontaneous transition of the transporter to the outward-open conformation and unbinding of the K⁺ ions.

This is corroborated by mutational studies of Asn237 and Tyr118 residues that act as gates in the inward-occluded conformation.

Additionally, the cooperativity in c-di-AMP binding is suggested, based on MD simulations and experiments.

Overall, the manuscript presents an interesting story, and the proposed mechanism of inhibition of KimA by c-di-AMP is plausible and supported by the data. I have however a number of points that should be resolved before the publication of the manuscript. I will focus mostly on MD simulations

Thank you for your very helpful suggestions.

1. The investigation of the ‘cooperativity’ or ‘priming’ of one binding site by c-di-AMP binding to the other is not very convincing. The authors say that ‘RMSF of all these residues were significantly reduced compared to the apo-protein..’ based on the table given in Fig 3 b. This table includes only averaged RMSF values from simulations with some (which?) error estimate. Closer inspection reveals that some of these values overlap within their error bars (1st row vs 2nd/3rd): Ile569, Phe565 and Phe568 in multiple instances do not seem to differ (within the error estimates provided). If the authors wanna say something about significance, I suggest showing individual data points from each simulations, and calculating p values for each compared pair. Visually, that would be better seen on a scatter plot, rather than in a table with many values.

We agree that the representation of this data was a little unclear. We have now restructured the data into a bar plot including all data points (Figure 3b). To allow p-tests to be run, we have pooled the data for the forward and reverse directions (which are symmetrical). We have updated the text and legend to match. The raw data, along with the “asymmetric” ligand state data, have now been moved to Supplementary Figure 3.

Further, it is not clear how the simulations without the ligand (apo state) were initiated: was c-di-AMP deleted from its binding site or did the authors used they previously published structure (ref 25?). A removal of a ligand from an existing structure would naturally cause large fluctuations of residues involving in its binding; it would be therefore of interest to compare the RMSF values with those from previous simulations of a ‘true’ apo conformation.

To run the apo states simulations, the ligand was indeed deleted from the binding site, followed by 30 ns of equilibrium with backbone positional restraints. We have clarified this in the methods section. It is worth pointing out that some of the most affected residues (e.g. Trp 592) are quite far from the ligand binding site.

Comparison with the previous apo state is difficult, as the limited resolution in this region did not allow unambiguous modelling of the sidechains. In addition, we are unsure of the ligand occupancy in the previous apo data, and it is possible that the data contained both ligand bound and apo states (as c-di-AMP was added to the sample prior grid preparation). Therefore, we have avoided direct comparison on these data.

2. The observation that KimA spontaneously transitions toward the inward-open conformation after removing (deleting?) c-di-AMP is very interesting, and deserves some additional analysis in my opinion. For example, the size of the intracellular opening, its water accessibility and potassium unbinding pathway(s) should be analyzed.

We agree that this is an important point, and we now include some structural analyses to the supplementary figures. These include visual comparison of post simulation state cavities using HOLE, quantification of the cavity size over the simulations using HOLE, visual inspection of the waters in the cavities at the end of the simulation, and comparison of Tyr118-Asn237 distance to water number. These are included in Supplementary Figure 7.

It would be also very interesting to put this transition into a broader perspective - since KimA has the popular LeuT fold, I'd assume that structures of related transporters in the inward-open conformation might exist. A comparison of such structures would be therefore very interesting and could deepen our understanding not only of KimA, but also of other proteins.

This is a good idea, and we have performed a structural comparison on the inward-open structure of BasC, which matched the post simulation no-ligand KimA snapshot using a FoldSeek search (Supplementary Figure 8). We also made a homology model of KimA using the BasC structure to assist in the comparison.

Further, in a similar spirit to the previous comment, can the authors clarify why this transitions occurs in current simulations, but did not seem to occur in previous MD simulations (ref 25), although both proteins are in the apo conformation?

As stated above, unfortunately we are unable to compare our data to that of the previous structure, owing to a limited reliability of the density in the cytoplasmic region and the uncertainty of the ligand occupancy in this state.

3. The PCA analysis could be improved. It is not clear whether the PCA was done on all trajectories together, or those with and without ligand separately. If the latter, then the meaning of eigenvectors 1 is different between these two sets (it is a different vector). It is also not clear what is the color code in the Supplementary Figure 10. I'd recommend performing PCA on all trajectories concatenated together, which should show clear basins corresponding to the 'inward-open' and 'inward-occluded' states.

We have now run PCA analysis on both the ligand bound and ligand free trajectories combined. We present the data for eigenvector 1 in a new main figure (Figure 5c), and for eigenvector 2 as Supplementary Figure 6. This doesn't change the overall findings for the PCA, but we agree that it allows for a more direct comparison between the ligand and no-ligand states.

4. The behavior of the used mutant R337A seems to be complicated: based on biochemical data, the authors conclude that R337 is important for high affinity binding of c-di-AMP, but its not necessary required for communicating c-di-AMP binding to the TMD, because the mutation reduced K⁺ transport (induced by c-di-AMP) by 44%. However, MD simulations suggest that, in terms of loop dynamics, R337A behaves similarly to the ligand-bound protein (Fig 5 a). Therefore, is it possible that R337 is (also) involved in the communication between c-di-AMP binding site and TMD, but the alanine mutation mimics (to some extend) R337 reduced dynamics upon c-di-AMP binding? Do R337A MD simulations also show the transition to the inward-open conformation?

Based on your comments we revisited the R337A simulations and decided to remove them from the manuscript. These simulations were initially run to look at the dynamics of the R337 loop, for which 3 x 500 ns were likely sufficient sampling. The sampling is likely too limited, however, to infer anything about more global structural dynamics, such as switching from

outward to inward conformations. In addition, we fear that the mutation might have structural side effects that are not reflected in the simulations.

Instead, we have toned down our conclusion on the role of the loop as a whole and on R337 in particular. There is still strong evidence (transport data, MD simulations in the presence and absence of c-di-AMP) that the loop gets stabilized in the presence of c-di-AMP. However, we cannot exclude that R337 has a major role in the communication and that the alanine mutation mimics to some extent the stabilizing function of the Arg-to-c-di-AMP interaction.

5. If the reduced flexibility of the loop is the underlying reason for c-di-AMP mediated inhibition of KimA, increasing its flexibility, e.g. via glycine mutations, should abolish the effect of c-di-AMP. Can the author at least discuss it (ideally perform such experiments, but I won't push for it).

We agree with the reviewer that additional mutations (e.g., to glycines) could be added to the loop in KimA_{R337A} to increase its flexibility. This could ultimately abolish inhibition by c-di-AMP despite its binding. However, this could also have unpredictable side effects on the stability of the protein, affecting the activity of KimA in unforeseen ways, as observed by us in other proteins. Following the argumentation from above (your point 4), we decided to not include the mutation in an MD simulation. Since we toned down our conclusion on the role of the loop in general, we also did not include a further discussion on possible mutations in our manuscript but believe that this authors' response is a better place for the discussion.

Reviewer #4 (Remarks to the Author):

The authors have presented a multidisciplinary study on the effect of c-di-AMP on the structural conformation and dynamics of a KUP transporter, KimA, using a combination of cryoEM, molecular dynamics, and functional assays. Overall, the data supports the authors interpretation and I have no major concerns. The two cryoEM structures presented are of good quality and the secondary structure appropriately modelled. The ligand density has also allowed the authors to place c-di-AMP into a suitable pose. However, there are a few areas in which the manuscript and structures could potentially be improved, listed below are some minor concerns and suggestions. Issues in modeling below should be corrected in order to uphold the rigor and quality expected in the field, but I note that they have not affected the interpretations made by the authors and therefore I would categorize them as minor revisions with regards to reviewing the manuscript as a whole.

-In PDB:8B70/EMD15894 (DDM map), residue R462, near the c-di-AMP binding site, is modeled outside the EM density and should be remodeled into the continuous density extending towards c-di-AMP.

Well spotted, thank you. We remodelled the side chain.

-Although the use of C2 symmetry in the final 3D refinement may result in a higher calculated global resolution, it may be concealing subtle asymmetric differences between the monomers. This is particularly salient for the present study as the authors note a level of cooperativity in c-di-AMP binding. The authors should carefully review their C1 generated maps to be sure they indeed have symmetry before forcing C2 symmetry during processing. It may be worth commenting within the manuscript that C1 displays no asymmetry, or if they find that it does then elaborate on their findings.

We had checked the C1 maps before applying the symmetry, no deviations from symmetry were found. A statement has now been added to the Methods section (lines 668ff).

-There are several residues (e.g., K466, E511, K402, N244, E521, K402, R573 in PDB:8B71, and K515, D509, N244 in PDB:8B70) on the periphery that do not have sufficient density to justify modeling the sidechains. The authors should review their structures and stub side chains outside of the EM density.

Thanks for the critical reviewing. We removed the side chains from the pdb file.

-Please make note of the residues shown in the supplemental videos (V1-V3) by either directly labeling within the video, identifying by color in the Supplementary Materials legend, or by remaking the video to be in a similar vantage point as figure 3a and noting in the V1-3 legends that the region is viewed as in figure 3a.

We have included the color code of the videos (basic=blue, hydrophobic=yellow, aromatic=orange, polar=green) in Fig. 3a.

-Add a panel to the supplement to show the cryoEM map-to-model fit of c-di-AMP and surrounding residues, as well as for the K⁺ site highlighted in figure 2.

We added these two panels as a new Supplementary Figure 2.

-Two of the modeled DDM in PDB:8B70 at the dimer interface (B/903, A/1003) do not have EM density to support their modeling, their inclusion is unnecessary, and they should be removed.

We have removed these two DDM molecules and also fitted two others better to the density.

-The summary schematic of figure 6 is a bit confusing. I suspect the authors were attempting a schematic similar to the one shown in the SMALP paper (ref 25), but the disjunction of the TM and CD domains makes the schematic unclear, especially in absence of labels of the structural elements/domains.

We have modified the schematic to align with the color scheme of figs 5 and 6. TM and CD are in fact not connected as they belong to different protomers. We have highlighted this by including a schematic of the KimA dimer in new panel A.

-In supplementary fig. 3 the authors may want to consider adding the moving average of the traces to the graphs.

We have now added this.

Reviewers' Comments:

Reviewer #3:

Remarks to the Author:

The authors have addressed all my comments. I recommend the paper for publication and I'd like to congratulate the authors a very nice story.